# Research

cognition/psychology

nonverbal vocalizations, context, emotion, nonlinear vocal phenomena, acoustic communication

**Author for correspondence:**
Andrey Anikin
e-mail: andrey.anikin@lucs.lu.se

# Do nonlinear vocal phenomena signal negative valence or high emotion intensity?

Andrey Anikin[1,2], Katarzyna Pisanski[2] and David Reby[2]

[1]Division of Cognitive Science, Lund University, Lund, Sweden
[2]Equipe de Neuro-Ethologie Sensorielle (ENES) / Centre de Recherche en Neurosciences de Lyon (CRNL), University of Lyon/Saint-Etienne, CNRS UMR5292, INSERM UMR_S 1028, Saint-Etienne, France

 AA, 0000-0002-1250-8261; KP, 0000-0003-0992-2477;
DR, 0000-0001-9261-1711

Nonlinear vocal phenomena (NLPs) are commonly reported in animal calls and, increasingly, in human vocalizations. These perceptually harsh and chaotic voice features function to attract attention and convey urgency, but they may also signal aversive states. To test whether NLPs enhance the perception of negative affect or only signal high arousal, we added subharmonics, sidebands or deterministic chaos to 48 synthetic human nonverbal vocalizations of ambiguous valence: gasps of fright/surprise, moans of pain/pleasure, roars of frustration/ achievement and screams of fear/delight. In playback experiments ($N = 900$ listeners), we compared their perceived valence and emotion intensity in positive or negative contexts or in the absence of any contextual cues. Primarily, NLPs increased the perceived aversiveness of vocalizations regardless of context. To a smaller extent, they also increased the perceived emotion intensity, particularly when the context was negative or absent. However, NLPs also enhanced the perceived intensity of roars of achievement, indicating that their effects can generalize to positive emotions. In sum, a harsh voice with NLPs strongly tips the balance towards negative emotions when a vocalization is ambiguous, but with sufficiently informative contextual cues, NLPs may be re-evaluated as expressions of intense positive affect, underlining the importance of context in nonverbal communication.

# 1. Introduction

In this study, we explore an understudied yet highly biologically and socially relevant set of voice features—nonlinear vocal phenomena (NLPs), which contribute to a harsh or rough voice

**Figure 1.** A synthetic yell of achievement (*a,b,c*) and moan of pleasure (*d,e,f*) resynthesized with and without NLPs. Subharmonics are characterized by the presence of both the main tone and a lower frequency at an integer fraction of the fundamental frequency or $f_0$. Relatively low-frequency amplitude modulation, often caused by supraglottal oscillators such as mucosal folds, produces closely spaced additional harmonics or sidebands around each harmonic of $f_0$. Deterministic chaos is marked by a characteristic 'noisy' appearance on a spectrogram and a very harsh perceptual quality. Observe that breathing (turbulent noise) looks superficially similar to chaos on the spectrograms. The moan also contains a discontinuity in $f_0$ contour (frequency jump) at approximately 3.2 s. See electronic supplementary material for more examples with audio.

quality—and use context manipulation as a methodological tool to test competing theories of the communicative significance of NLPs in human nonverbal vocalizations such as moans, roars and screams.

Mechanistically, NLPs result from perturbations in the typical rhythmic vibration of the vocal folds that cause deviations from regular, tonal voice production (see [1–4] for reviews). This can result in frequency jumps, subharmonics, sidebands and deterministic chaos (figure 1), which give the voice a perceptual quality of harshness, roughness or instability. It is increasingly recognized that NLPs are not simply by-products of vocal production, but rather have the potential to convey important information about the caller. Yet, while NLPs have received attention from researchers studying the evolution and functions of animal calls [5–9], the nature of this information, and thus the function of NLPs in human and animal vocalizations, remains debated. NLPs primarily occur when $f_0$ or subglottal pressure are high, which is indicative of high vocal effort and an aroused physiological state [5,10]. In addition, NLPs render a signal less predictable, which attracts attention [7] and reduces habituation in listeners [8]. Accordingly, they are often treated as an expression of high arousal or emotion intensity [6,7,11]. On the other hand, a rough voice often signals aggression [2,6,11–15] and other negative emotions [16,17] in line with Morton's motivation-structural rules [18]. Recent research has also shown that rapid spectrotemporal modulation correlates with the aversiveness of artificial click trains [19] and with the perceived fear intensity in human screams [12], suggesting that listeners may experience both amplitude modulation (sidebands) and chaos as unpleasant.

**Table 1.** Acoustic descriptives of nonverbal vocalizations: mean [range across stimuli].

| call type | N (F/M) | duration (ms) | pitch (Hz) | HNR[a] (dB) | roughness[b] (%) |
|---|---|---|---|---|---|
| gasps | 7 (5/2) | 883 [500, 1449] | 477 [236, 946] | 10 [6, 16] | 22 [15, 27] |
| | | | | 8 [3, 14] | 25 [16, 32] |
| | | | | 5 [1, 10] | 33 [25, 36] |
| moans | 26 (14/12) | 1009 [437, 3843] | 386 [116, 848] | 13 [4, 21] | 14 [5, 34] |
| | | | | 10 [2, 18] | 19 [9, 34] |
| | | | | 7 [1, 17] | 29 [17, 38] |
| roars | 7 (0/7) | 994 [412, 1540] | 413 [286, 753] | 16 [15, 19] | 9 [4, 15] |
| | | | | 12 [8, 16] | 19 [7, 28] |
| | | | | 4 [−1, 8] | 37 [31, 43] |
| screams | 8 (7/1) | 1273 [275, 2451] | 1243 [596, 1889] | 21 [15, 25] | 13 [7, 26] |
| | | | | 18 [14, 23] | 17 [11, 27] |
| | | | | 15 [8, 17] | 27 [16, 37] |

[a]HNR (harmonics-to-noise ratio) for the three manipulations: no NLPs (top value), subharmonics/sidebands (centre), and chaos (bottom).
[b]Roughness: the proportion of the modulation spectrum that falls within 30–150 Hz.

An important limitation of past work is that most studies report only general measures of voice quality such as perturbations in frequency (jitter) and amplitude (shimmer), the amount of energy in harmonics (harmonics-to-noise ratio, HNR) or vocal roughness (the amount of spectrotemporal modulation in the 'roughness' frequency range [20]). Few studies have targeted NLPs; yet, different NLPs carry specific information that is not fully captured by general acoustic measures such as HNR [16]. It is therefore vital to supplement general spectral descriptives with a more fine-grained analysis of NLPs. This is technically challenging but feasible; for instance, several papers have shown that NLPs are involved in conveying intense distress in the pain vocalizations of human infants [13,14] and adult actors [15]. Here, we use a novel acoustic analysis and resynthesis software, *soundgen* [21], to experimentally add NLPs to human nonverbal vocalizations.

Using this innovative methodology, we empirically test two accounts of the role of NLPs in the perception of vocal affective signals: (i) NLPs are non-specific and attention-grabbing indicators of high arousal, and thus not necessary specific to negative emotional and motivational states, or (ii) NLPs preferentially convey negative affect. In an earlier study [16], we manipulated NLPs in synthetic nonverbal vocalizations and observed a surprisingly limited effect on listeners' ratings of arousal. In contrast, adding NLPs (especially chaos) to vocalizations shifted their interpretation towards unpleasant emotions. Unfortunately, those results proved inconclusive for laughs—the only call type that was consistently associated with positive valence ratings [16]. In other words, because NLPs are often found in calls that express aversive states, the effects of NLPs on emotion intensity and negative valence are hard to tease apart. To address this confound, here we control the perceived valence of vocalizations and experimentally add NLPs to systematically test their influence when context cues are positive, negative or absent. If NLPs primarily signal high arousal, their addition should boost the perceived intensity of emotion in both positive and negative contexts. Conversely, if NLPs convey only negative affect, their presence should boost the perceived emotion intensity in negative contexts only and attenuate it in positive contexts.

To test this idea, we exploited the highly ambiguous nature of many human nonverbal vocalizations, which can be perceived by listeners as either positive or negative, depending on contextual cues. Having pre-tested a variety of vocalizations to select ambiguous exemplars, we resynthesized the vocalizations either with or without NLPs. Independent samples of participants then assessed the valence and intensity of these stimuli in the absence of contextual cues to test the effectiveness of context manipulation (Experiment 1) and, crucially, in a positive or negative context created by short vignettes and visual illustrations (Experiments 2 and 3). To make our findings more generalizable, in this study, we tested two types of NLPs (subharmonics/sidebands and chaos) and four types of nonverbal vocalizations: screams, roars, gasps and moans (table 1).

# 2. Material and methods

## 2.1. Stimuli

Vocal stimuli derived from 48 nonverbal vocalizations (screams, roars, gasps and moans) that were ambiguous in valence. These were taken from a published corpus of spontaneous, non-staged vocalizations [22].

Ambiguity was verified using several methods. Authentic vocalizations may be generally more ambiguous compared to acted portrayals of particular emotions [23]. The availability of intensity ratings from the validation study of 260 sounds [22] made it possible to identify an initial set of 61 screams, roars, gasps and moans, all of which obtained high ratings on one or more positive and negative emotions. This initial selection was then validated in a pilot study, in which 25 participants were asked to rate the apparent intensity of either a positive or negative emotion for each of four call types, or else to skip the sound as unsuitable for this emotion. The 61 vocalizations in this pilot study were fully synthetic, but they were designed to be as similar as possible to the original recording and contained the same NLPs, if any. Long bouts were truncated to be no more than 3.8 s in duration. Based on the results of this pilot study, 13 out of 61 sounds were excluded because (i) participants often chose the 'skip' option when rating them in either positive or negative scenarios, or (ii) there was a large difference in intensity ratings between the positive and negative scenarios.

## 2.2. Acoustic synthesis

The remaining 48 sufficiently ambiguous vocalizations were each synthesized in three versions: without any NLPs, with subharmonics and/or sidebands, and with chaos (simulated with jitter, rapid random fluctuations of $f_0$; figure 1). This resulted in a total of 144 fully synthetic vocal stimuli that were based on the 48 original recordings taken from [22], which were used as templates for synthesis.

The synthesis and acoustic analyses were performed with *soundgen*—a parametric voice synthesizer previously validated as a tool for synthesizing human nonverbal vocalizations [21] and for manipulating NLPs [16]. With this approach, we managed to create fully controlled yet highly realistic synthetic copies of the original vocalizations. The main acoustic characteristics of vocal stimuli are shown in table 1. Stimulus duration and pitch were similar across the three manipulated versions. Harmonics-to-noise ratio (HNR) and roughness varied across the three manipulated versions as predicted. Thus, for all four call types, vocalizations with added chaos were on average the noisiest (lowest HNRs, highest roughness), vocalizations with subharmonics/sidebands had intermediate HNRs and roughness, and versions without any NLPs were the most tonal (highest HNRs, lowest roughness; table 1). To avoid creating artificial-sounding stimuli, we based our synthesis of NLPs on comparable real-life recordings [22]. Accordingly, the duration and intensity of subharmonics, sidebands and chaos varied across stimuli and was considerably greater in screams, and especially in roars, compared to moans and gasps.

## 2.3. Procedure

We performed three playback experiments on independent samples of a total of 900 participants. Figure 2 illustrates the design of each experiment. The valence ratings obtained in neutral, positive and negative contexts (Experiments 1 and 2) enabled us to verify the effectiveness of contextual cues in shifting the perception of these ambiguous vocalizations, as could be predicted based on previous research [23,24]. We were particularly interested in confirming the finding by Atias *et al*. [23] that contextual cues sometimes suffice not only to modulate, but even to reverse or 'flip' the perceived valence of a vocalization. In addition, Experiments 1 and 2 addressed the effect of NLPs on perceived valence. The intensity ratings obtained in positive and negative contexts (Experiment 3) made it possible to test whether the addition of NLPs boosted the intensity of any emotion (as predicted by the attention-grabbing account) or specifically increased the perceived intensity of only negative affective states (as predicted by the harsh-is-aversive account).

### 2.3.1. Experiment 1: valence and intensity ratings, no context

Participants ($N = 300$) heard 48 vocal stimuli in four unmarked blocks (gasps, moans, roars and screams). The order of blocks and sounds within each block was randomized. For each vocal stimulus, one of three manipulated versions (without NLPs, with subharmonics/sidebands or with chaos) was drawn at random for each trial by a computer algorithm, ensuring that each participant heard only one version

| call type | experiment | scenario | | | |
|---|---|---|---|---|---|
| | | positive | | negative | |
| | | verbal cues* | visual cues | verbal cues* | visual cues |
| all | 1a | How do they feel? | – | How do they feel? | – |
| | 1b | How intense is the emotion they are experiencing? | | How intense is the emotion they are experiencing? | |
| gasps | 2 | What a pleasant surprise! How do they feel? | | What a nasty surprise! How do they feel? | |
| | 3 | What a pleasant surprise! How happy are they? | | What a nasty surprise! How frightened are they | |
| moans | 2 | They are having a massage/sex /tasty food! How do they feel? | | They are in pain! How do they feel? | |
| | 3 | They are having a massage/sex /tasty food! How much are they enjoying it? | | They are in pain! How much are they suffering? | |
| roars | 2 | They are watching a match, and their favourite team just scored! How do they feel? | | They are watching a match, and their favourite team team just lost! How do they feel? | |
| | 3 | They are watching a match, and their favourite team just scored! How happy are they? | | They are watching a match, and their favourite team team just lost! How frustrated are they? | |
| screams | 2 | They just won a lottery! How do they feel? | | They are being chased by a ghost! How do they feel? | |
| | 3 | They just won a lottery! How happy are they? | | They are being chased by a ghost! How frightened are they? | |

*The outcome measure was the rating on a horizontal visual analogue scale labeled as follows: experiments 1a and 2 'terrible...great'; experiment 1b "not at all intense... extremely intense"; experiment 3 "not at all... a lot" (with a skip button).

**Figure 2.** The design of Experiments 1–3.

of each stimulus. For half the participants, the outcome measure was the perceived valence of each sound, which raters indicated with a horizontal slider (How do they feel?) ranging from Terrible to Great (figure 2). The thumb was reset to the middle of the scale after each trial. The other half of participants rated the emotion intensity (How intense is the emotion they are experiencing?), ranging from Not at all intense to Extremely intense; the thumb returned to zero (far left) after each trial. Stimuli could be replayed as many times as desired, and there was no time limit for responding.

### 2.3.2. Experiment 2: valence ratings, positive or negative context

A second sample of participants ($N = 300$) were presented with the same fully randomized sounds in the same blocks as in Experiment 1 and used the same rating scale. However, here each block was accompanied by a short vignette and visual illustrations of a specific eliciting context, such as winning a lottery (where screams may indicate positive valence) or being chased (where screams may indicate negative valence). The images were obtained from the open collection at http://clipart-library.com/openclipart.html. Only the images shown in figure 2 were used, the same for all participants and in all trials in a block—that is, versions of the same prototype stimulus with and without NLPs were presented with the same (positive or negative) contextual cues, enabling us to test the within-stimulus effect of acoustic manipulation in positive and negative contexts.

### 2.3.3. Experiment 3: intensity ratings, positive or negative context

A third sample of participants ($N = 300$) were presented with the same sounds as in Experiments 1 and 2, and the set-up was the same as in Experiment 2, except that the verbal vignette indicated a specific emotion (figure 2 for details), and the rating scale measured its perceived intensity from Not at all to A lot. Being forced to rate the intensity of a single (positive or negative) emotion placed additional constraints on the possible responses compared to the first two experiments, in which the slider ranged from negative to positive valence. Accordingly, participants were also given the option to skip a trial if they found that the vocalization was unsuitable in the present context. This Skip button was used in approximately 15% of trials. The rating scale was reset to zero intensity (far left) after each trial.

## 2.4. Participants

Sample sizes were chosen prior to testing, aiming to have each of 144 vocal stimuli (48 prototypes × 3 manipulations) rated 50 times in each context (none/positive/negative) and on each scale (valence/intensity). In a previous study with similar sounds and rating scales [25], 50 ratings per vocal stimulus translated into good precision of estimates, with 95% credible intervals of approximately ±5% for

individual vocalizations or call types represented by several vocalizations. This is sufficient for detecting the expected effects of NLPs on perceived valence, which were previously reported to be 5–20% in magnitude [16]. Because each participant heard only one-third of the stimuli, 300 participants (50 × 3 NLP manipulations × 2 response scales) were recruited for Experiment 1. Experiments 2 and 3 had only one response scale but two conditions each (positive/negative scenarios), so 300 participants per experiment again translated into 50 ratings per vocalization and condition.

A total of 900 participants, fluent in English and with no self-reported hearing problems, were thus enrolled in the three experiments on the online recruitment platform Prolific. To ensure reliable data, responses from seven participants (less than 1% of data) were excluded as clearly invalid (e.g. participants reported technical problems or had very rapid identical responses to many consecutive vocalizations). The remaining 893 participants, all of whom had completed at least 30 out of 48 trials, were included in the analysis. In this sample, 411 (46%) self-identified as female, 471 (53%) as male and 9 (1%) as 'other'. According to the statistics on the website, at the time of testing, approximately 75% of participants on this platform were native English speakers, and 72% were between 20 and 40 years of age (https://www.prolific.co/demographics).

## 2.5. Data analysis

The original dataset, scripts for acoustic analysis, audio stimuli and html files for running the experiments are available in the electronic supplementary material and from http://cogsci.se/publications.html.

Data from all three experiments (40 978 trials after excluding the 15% of *Skip* responses in Experiment 3) were analysed without aggregation, in a single multilevel model with three interacting main effects: call type (roar/scream/moan/gasp), measure (valence/valence in a positive context/valence in a negative context/intensity in a positive context/intensity in a negative context) and manipulation (no NLPs/subharmonics or sidebands/chaos). The outcome variable was the rating of a sound on a continuous scale, modelled with zero-one-inflated beta distribution [26] with four separately modelled parameters: (i) *mu*: the mean of beta distribution capturing non-extreme responses between 0 and 1; (ii) *phi*: the precision of beta distribution; (iii) *zoi*: zero-one inflation, the probability of answering 0 or 1 rather than a number in the interval (0, 1); (iv) *coi*: conditional one inflation, the probability of answering 1 rather than 0.

To make the model computationally tractable, its structure was slightly simplified relative to modelling all four parts with the full list of fixed and random effects. Model selection was performed on a subset of data using *WAIC*, Watanabe–Akaike information criterion [27], which suggested that modelling *phi* did not improve the model's predictive power. This part was therefore omitted, together with the random effects in the models of *zoi* and *coi*. The model of *mu* contained a random intercept for each participant and each unique vocal stimulus ($N = 144$, 48 prototypes × 3 manipulations). In addition, the effect of adding NLPs in each experiment was allowed to vary across stimuli by specifying the corresponding random slopes for each of 48 prototype sounds. All models were fit using the R package *brms* [28] with conservative priors. Posterior distributions of model parameters were summarized by their medians and 95% credible intervals, reported in the text as 95% CIs (on the interpretation of Bayesian CIs, see [29]). In the text below all responses are converted to a scale of [0, 100] and marked with the percentage sign (%) as a reminder of the range of values. The effect sizes that are reported as percentages thus correspond to the relative change in slider position on the continuous response scale and should not be confused with proportional change.

To test for inter-rater reliability, we calculated the correlation between the responses of each participant and the average ratings provided by all participants in each of the three experiments, as well as the intraclass correlation coefficient (*ICC*). Both measures demonstrated moderate agreement when the stimuli were played back within a particular context ($r = 0.65$ and 0.61 for valence and intensity, respectively; $ICC = 0.41$, 95% CI [0.37, 0.45] and 0.30 [0.26, 0.34], which is comparable to previous studies using similar stimuli [16,21,25]. When stimuli were presented without any context, the intensity ratings were again moderately consistent ($r = 0.66$, $ICC = 0.36$ [0.31, 0.42]), but the valence ratings became considerably less stable ($r = 0.50$, $ICC = 0.20$ [0.16, 0.25]), which was expected because the stimuli were specifically selected to be ambiguous in terms of their valence. Adding interactions with the listener's gender to the main model failed to improve its predictive power (difference in $WAIC = 5.0$, $SE = 18.0$ in favour of the simpler model). Accordingly, below we present the results for male and female raters together.

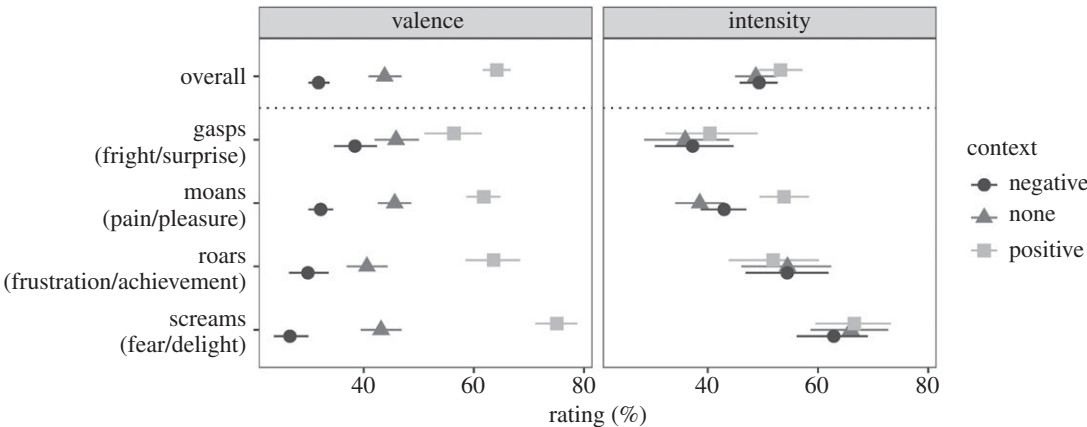

**Figure 3.** The effect of context on the perceived valence and emotion intensity of nonverbal vocalizations averaging over all three manipulated versions (no NLPs, subharmonics and chaos). Medians of posterior distributions and 95% CIs.

# 3. Results

## 3.1. Context

Confirming the effectiveness of context manipulation, the scenarios in which vocalizations were presented had a large impact on their perceived valence. Averaging across vocal stimuli with and without NLPs, the vocalizations were rated as neutral or slightly negative in the absence of any contextual cues, demonstrating their inherent ambiguity (figure 3, left panel). Providing participants with verbal and visual contextual cues reversed or 'flipped' the perceived valence of most vocalizations (e.g. from positive to negative or vice versa), for all four call types and for all 148 unique vocalizations (143/148 based on observed averages instead of fitted values). For example, screams were on average rated as an expression of a pleasant emotion in the context of winning a lottery (valence = 75.1% on a scale from 0 to 100%, 95% CI [71.2, 78.8]), but these same screams were predominately rated as aversive in the context of being chased by a ghost (valence = 26.6% [23.8, 29.7]), a difference in the rated valence of 48.4 percentage points, 95% CI [43.3, 53.1]. Likewise, other call types were rated as considerably more aversive in the negative-context versus positive-context condition: a difference of 33.8% [27.4, 39.9] for roars, 29.7% [26, 33.3] for moans and 17.9% [10.8, 24.8] for gasps.

In comparison, context had a relatively muted effect on the perceived intensity of affective vocalizations (figure 3, right panel), demonstrating that the presented stimuli fit in well with both the positive and the negative scenarios. For example, for screams, the average intensity of delight was only 4.0% [−1.1, 9.0] above the average intensity of fear. The effect of context on the perceived emotion intensity was also too small and uncertain to detect for gasps (a difference of 3.1% [−2.7, 9.5]) and for roars (a difference of 2.3% [−8.3, 4.0]). Interestingly, however, moans were consistently rated as more intense in a positive than negative context (a difference of 10.9% [7.2, 14.5]), indicating that the same moan could be perceived as either mild pain or great pleasure.

In sum, contextual manipulations effectively controlled whether ambiguous vocalizations were predominantly perceived as either positive or negative in valence, without an undue effect on their perceived intensity (except for moans). The tested call types also covered a wide variety of perceived intensity levels, with screams perceived as the most intense, and gasps the least intense.

## 3.2. Nonlinear vocal phenomena

As expected, adding subharmonics/sidebands and particularly deterministic chaos (see the spectrograms in figure 1) caused all vocalizations to be perceived as more negative in valence (aversive) when presented without any contextual cues (figure 4, top left panel). Averaging across all four call types, adding subharmonics/sidebands made a vocalization 6.9% [4.5, 9.2] more aversive and adding chaos made it 15.0% [12.8, 17.2] more aversive, relative to the same vocalization without NLPs. Adding subharmonics/sidebands to moans had the smallest effect, making them only negligibly more aversive (2.9%, 95% CI [0.8, 5.1]), whereas adding chaos to roars had the largest effect, making roars sound 21.0% [16.4, 26.0] more aversive compared to roars without added NLPs. Notably, in many cases the effect of NLPs was not merely a matter of making an already aversive vocalization more negative-sounding, but rather, of

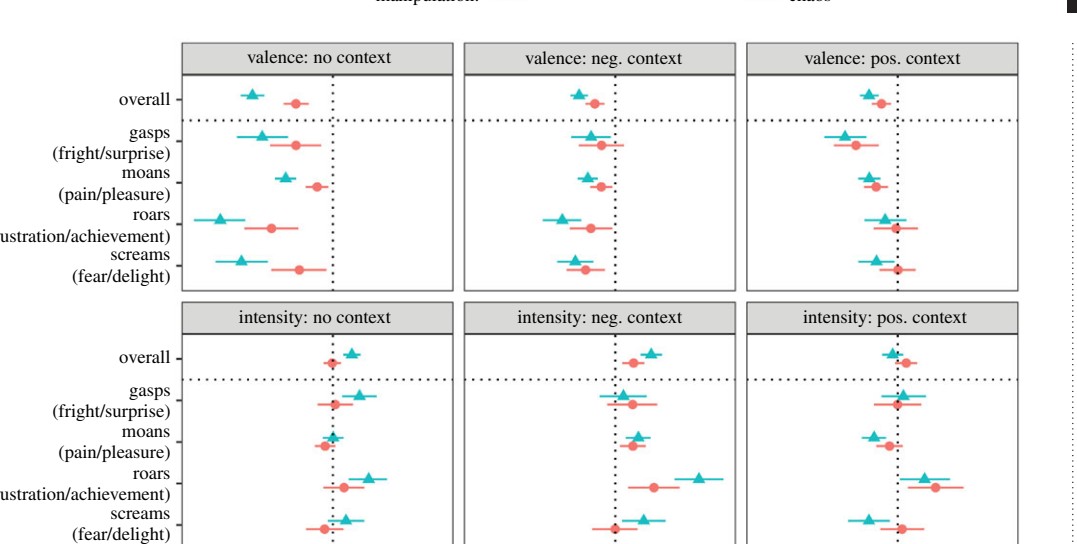

**Figure 4.** The perceptual effect of adding NLPs to nonverbal vocalizations as a function of call type, context and the outcome measure (valence or emotion intensity). Medians of posterior distributions and 95% CIs.

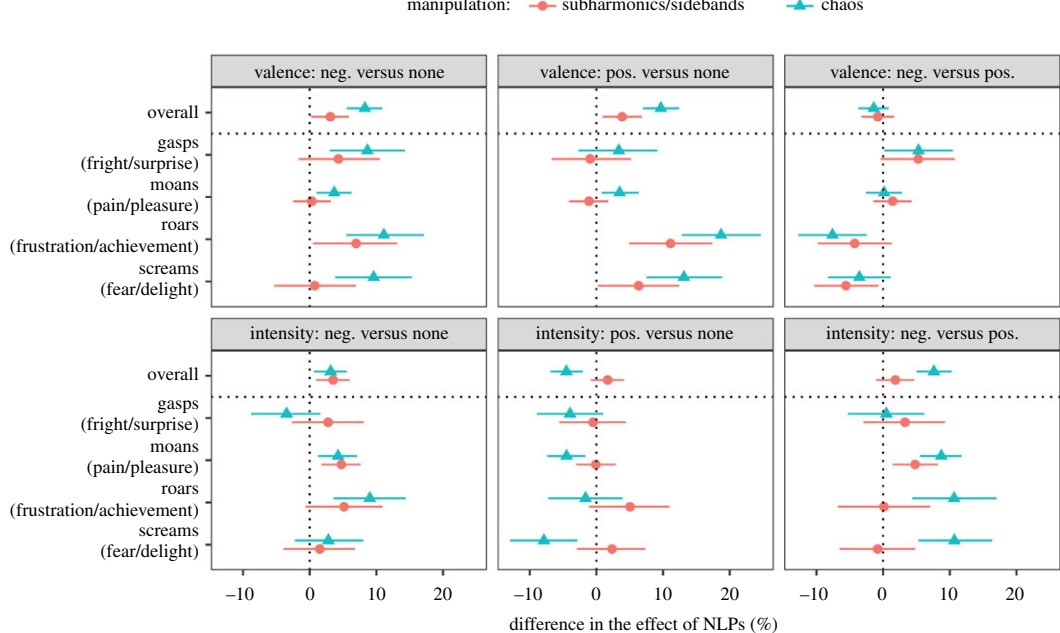

**Figure 5.** Context-dependent differences in the perceptual effect of adding NLPs to nonverbal vocalizations. These are contrasts between the effects shown in figure 4. For example, 'valence: neg. versus none' shows the difference in the effect of NLPs on valence in a negative context versus in the absence of contextual cues. Medians of posterior distributions and 95% CIs.

reversing its perceived valence: 17 out of 48 vocalizations (16/48 based on observed instead of fitted values) were on average perceived as positive without NLPs and as negative with NLPs, including 4/7 roars, 5/8 screams, 6/26 moans and 2/7 gasps.

When vocalizations were presented in a positive or negative context, the addition of NLPs again made the perceived valence more negative, but not as much as in the no-context condition (figure 4, top panel). Thus, the aversive effect of chaos dropped from 15.0% [12.8, 17.2] in the no-context condition to 6.8% [5.1, 8.4] in the negative-context condition and to 5.4% [3.6, 7.0] in the positive-context condition, corresponding to a reduction of 8.3% [5.6, 10.9] and 9.7% [7.0, 12.4], respectively (figure 5, top panel). This is unlikely to be the result of a ceiling effect because average valence ratings did not approach the extreme values of 0 or 100%

even in the presence of contextual cues. Instead, it appears that listeners rating valence paid less attention to voice quality cues when the pleasant or unpleasant nature of the experienced emotion was known *a priori*.

Notably, there was no overall difference in the aversiveness of NLPs in negative versus positive contexts (−0.8% [−3.3, 1.6] for subharmonics/sidebands, −1.4% [−3.7, 0.8] for chaos; figure 5, top panel), suggesting that NLPs were not reinterpreted as an expression of intense positive affect when the emotion was known to be pleasant. This is the result that would be expected if NLPs always convey aversiveness rather than emotion intensity. On the other hand, in the most intense calls with the strongest NLP manipulations (roars), NLPs no longer had a detectable effect on valence when vocalizations were presented in a positive context (figure 4, top panel), as if two perceptual effects of NLPs (aversiveness and high intensity) were brought in opposition in the positive-context condition and cancelled each other out.

If NLPs were always aversive, we would expect them to boost the perceived emotion intensity in the negative-context condition, but to attenuate it in the positive-context condition (figure 2). What we actually observed (figure 4, lower panel) was a weak positive effect of NLPs on intensity ratings in the negative-context condition (3.4% [1.3, 5.5] for subharmonics/sidebands, 6.7% [4.7, 8.7] for chaos) and no overall effect of NLPs in the positive-context condition (1.6% [−0.5, 3.6] and −0.9% [−2.9, 1.0], respectively). This difference, while relatively small, was robust statistically: the addition of chaos enhanced the perceived emotion intensity by an additional 7.6% [5.0, 10.3] when this emotion was negative rather than positive (figure 5, bottom panel). Unexpectedly, and in marked contrast to the large effect of NLPs on valence ratings, the effect of NLPs on emotion intensity in the no-context condition was absent for subharmonics/sidebands (−0.1% [−1.7, 1.5]) and quite modest for chaos (3.6% [1.9, 5.2]), further weakening the hypothesis that NLPs are salient markers of urgency or high arousal levels. However, once again we observed revealing differences between call types, particularly in the positive-context condition. Screams and moans displayed a clear aversive effect of NLPs: boosted perceived levels of fear and pain and attenuated levels of delight and pleasure (figure 4, lower panel). In roars, on the contrary, the addition of NLPs noticeably increased the perceived intensity of both achievement (subharmonics/sidebands by 7.1% [1.9, 12.3], chaos by 5.0% [0.4, 9.8]) and frustration (subharmonics/sidebands by 7.2% [2.4, 12.0], chaos by 15.6% [11.0, 20.2]). In other words, roar-like yells could express both positive (achievement) and negative (frustration) affect, and in either case adding NLPs enhanced the perceived intensity of emotion. The effects of NLPs in gasps, and of subharmonics/sidebands in screams, were less consistent (figure 4, lower panel).

Participants rating the intensity of vocalizations in Experiment 3 could skip a trial if they felt that the given vocalization was not well suited to the provided scenario. We analysed how often they did so as a function of the context and type of NLP added to the vocalization. The main finding was that a harsh voice quality in any type of vocalization was considered to be more naturally compatible with a negative rather than positive context, particularly for chaos. In a negative context, the addition of NLPs slightly decreased the probability of skipping the sound as unsuitable (by 2.8% [0.8, 5.1] for subharmonics/sidebands and 3.1% [0.1, 11.6] for chaos, averaging across call types). NLPs thus fit in well in a negative context, and it was rather their absence that was experienced as unnatural. In a positive context, the addition of subharmonics/sidebands had no effect on the probability of the vocalization being skipped (a difference of 0.3% [−1.8, 2.5]). However, adding chaos made a vocalization 5.1% [1.0, 12.9] more likely to be deemed unsuitable as an expression of a positive emotion. The model without an interaction with call types was preferred by *WAIC* (difference = 6.3, *SE* = 4.4), indicating that this expectation to find NLPs in aversive calls generalizes to all tested call types.

## 4. Discussion

We investigated the communicative significance of a harsh or rough voice quality caused by nonlinear phenomena (NLPs) in human nonverbal vocalizations (gasps, moans, roars and screams), which were synthesized in three versions: without NLPs, with controlled amounts of subharmonics and sidebands (growl-like voice quality), or with deterministic chaos (harsh voice quality). By focusing on exemplars of ambiguous human vocalizations and presenting their resynthesized variants in various contexts (positive, negative or no context), we were able to test whether the presence of NLPs (i) enhances the intensity of any perceived emotion, whether positive or negative, or (ii) shifts the interpretation towards negative emotional states. We observed convincing support for the second hypothesis, but also some evidence consistent with the first hypothesis, suggesting that they are not mutually exclusive and that both effects of NLPs may coexist.

Our main finding is that listeners strongly associate NLPs with aversive states when the basic valence of the underlying emotion is in doubt. Although we only modified NLPs, preserving all other acoustic characteristics of the stimuli (duration, amplitude and spectral envelopes, fundamental and formant frequencies, etc.), this targeted modification of voice quality sufficed not merely to change, but to reverse the perceived valence of approximately one-third of the tested stimuli from positive to negative. In comparison, the potential of NLPs for conveying high emotion intensity was quite modest, albeit also clearly present. Because NLPs often occur in vocalizations that express unpleasant emotions such as fear and pain, these two perceptual effects appear to work in synergy, causing sounds with NLPs to be interpreted as a sign of a powerful and unpleasant emotion [13–16]. However, if contextual cues convince listeners that the emotion is pleasant, the two effects of NLPs appear to conflict, making the outcome less predictable. For instance, adding NLPs to vocalizations presented as moans of pleasure and screams of delight decreased the apparent intensity of these positive emotions, while the presence of NLPs in roars of achievement was accepted by listeners as a sign of intense achievement.

Following examples from natural recordings [22], we synthesized roars with particularly strong NLPs, as can be seen from the great drop in their harmonicity and rise in roughness (table 1). It is therefore particularly revealing that these profoundly harsh sounds were nevertheless judged as more positive than gasps and moans with relatively mild NLPs. Presumably, an assertive or aggressive attitude expressed by yells and roars of delighted sport fans (the scenario used in this study) is easier to reconcile with the generally aversive quality of NLPs [18], while their presence is jarring in more prosocial sounds such as moans of pleasure. Consistent with this explanation, we previously observed that listeners associated subharmonics with a dominant attitude and physical effort [16]. In this sense, a harsh voice is analogous to visible tears, which ordinarily convey sadness [30], but in the right context can also boost the perceived intensity of a compatible positive emotion such as ecstatic happiness or relief [31].

In other words, emotion intensity may be regarded as a general 'temperature knob' that increases the probability of activating the displays that are particularly extreme (e.g. screams) or relatively atypical in the present context (e.g. sobbing for joy). As a result, sensory cues such as tears and NLPs, which are normally associated with negative emotions, may also be triggered by intense positive affect. For instance, people scream not only when they are afraid or in pain, but also when they are extremely happy [23,32], and receivers appear to take this into account as they use the same probabilistic reasoning to infer the signaller's likely internal state.

Looking at specific types of NLPs, chaos was consistently perceived as more aversive than subharmonics/sidebands. Although in line with previous investigations [16] and the amount of accompanying change in HNR and roughness (table 1), this finding should be treated with caution. Because we aimed to make the manipulated stimuli as natural-sounding as possible, the version with subharmonics/sidebands also contained a small amount of jitter, while the 'chaos' version sometimes contained subharmonics, albeit perceptually less noticeable than chaos. The classification of manipulations into 'subharmonics/sidebands' and 'chaos' is thus not absolute, and in this study, our priority was to investigate the perceptual effects of vocal harshness in general rather than to contrast different types of NLPs. The observed difference between manipulations, however, is suggestive and merits further research.

It is perhaps worth reiterating that in real life, people do produce calls with NLPs in both negative and positive contexts, so the discovered association of NLPs with negative valence may represent a perceptual bias rather than a true property of vocal production. Likewise, animal communication provides many examples of nonhuman calls with NLPs that are not aversive, such as affiliative vocalizations of red wolves [6], pant-hoots of chimpanzees [9] and songs of humpback whales [5]. Accordingly, in bioacoustics, the most common interpretation is that NLPs signal increasing arousal [2,6,11] and help to capture the listeners' attention [7,8]. From this point of view, it is intriguing that human listeners so consistently associate deterministic chaos, and to a smaller degree subharmonics and sidebands, with negative affect or distress, while the same acoustic manipulations appear to have little effect on the perceived level of arousal [16] or emotion intensity (this study). It will also be interesting to investigate whether the harsh-is-aversive perceptual bias affects the way people perceive the vocalizations of other species, including farm animals and pets. If the same acoustic rules are applied in an inter-species context [33], people might have a tendency to interpret any noisy call with NLPs as a sign of distress or aggression, which may not always be correct.

The role of NLPs is an important issue in itself given the growing recognition of their abundance not only in animal vocalizations [2,6,9,11], but also in human nonverbal communication [13–16]. More broadly, however, this investigation is an example of how contextualizing sensory cues can contribute

to a deeper understanding of their evolutionary function and meaning, both in humans and in other animals. Context-dependent interpretation of vocalizations is not a uniquely human ability. For example, bonobos produce acoustically similar contest hoots in play and in real conflicts and interpret these signals flexibly, relying on the accompanying gestures, which are 'softer' in play [34]. Apart from such multimodal integration, the meaning of communicative displays is open to more general contextual cues. For example, while there is some meaningful acoustic variation in primate vocalizations [35,36], calls like screams [37] and grunts [38] occur in such a wide range of contexts that the meaning of each sound considered in isolation from its social context is rather vague. It is the listeners' capacity for flexible, context-sensitive interpretation of such ambiguous signals that enriches their communicative potential. Accordingly, it has been hypothesized that a graded repertoire with highly variable and mutually complementary visual and auditory signals would optimize information transfer in habitats with good visibility and low noise levels, as well as in close-range rather than long-range calls. By contrast, discrete displays with categorical perception may be more effective when contextual cues are absent or when the communication channel is noisy [39,40].

Like other apes, humans have a nonverbal vocal repertoire that includes many graded stimuli [32], which makes these signals considerably more expressive than the small number of distinct call types would suggest, but this also means that nonverbal vocalizations may sometimes rely on a rich, multimodal context for disambiguation. In this respect, it is noteworthy that manipulating the verbal and visual context in which vocalizations were embedded had a major effect on valence ratings. Furthermore, in line with the recent report by Atias *et al*. [23], even their basic affective state—positive or negative valence—was affected by context. Thus, the same vocalizations could effectively convey pleasure or pain, delight or fear, achievement or frustration, pleasant surprise or fright. Naturally, not every vocalization is so pluripotent: we focused specifically on the call types that can each express a range of emotions [32] and pre-selected the most ambiguous examples from each call type. Revealingly, the communicative impact of adding or enhancing particular acoustic features, such as NLPs [16] and a breathy voice quality [25], appears to be greater in ambiguous vocalizations, and even more so when listeners have no contextual information to guide them (Experiment 1 in this study). Presumably, even relatively subtle changes in the signal can produce large perceptual effects when uncertainty is high, while the same acoustic changes are largely ignored when the meaning of a signal is transparent. At the same time, there is some leeway in the meaning of all human vocalizations. For example, laughs are typically associated with amusement and are therefore less ambiguous than screams and moans, but they can still express a range of social attitudes including malice, sarcasm and nervousness [24,41,42]. Thus, contextual cues are likely to enrich the meaning of most nonverbal displays, even those that are not inherently ambiguous, a hypothesis that now needs to be empirically tested.

Ethics. The research was approved by the institutional ethics review board, *Comité d'Ethique du CHU de Saint-Etienne* (IRBN692019/CHUSTE).

Data accessibility. The datasets supporting this article have been uploaded as part of the electronic supplementary material.

Authors' contributions. A.A. conceived the study, collected and analysed the data and drafted the initial manuscript. A.A., K.P. and D.R. synthesized the stimuli, designed the experiments, revised and approved the final manuscript, and agree to be accountable for the work.

Competing interests. The authors report no competing interests.

Funding. A.A. was supported by internal funding from Lund University. K.P. and D.R. were supported by the University of Lyon IDEXLYON project as part of the 'Programme Investissements d'Avenir' (ANR-16-IDEX-0005) funding to D.R.

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
