## [Reviewer comments · Royal Society Open Science]

Review History

RSOS-201306.R0 (Original submission)

Review form: Reviewer 1

Is the manuscript scientifically sound in its present form?

Yes

Are the interpretations and conclusions justified by the results?

Yes

Is the language acceptable?

Yes

Do you have any ethical concerns with this paper?

No

Have you any concerns about statistical analyses in this paper?

No

Recommendation?

Accept as is

Comments to the Author(s)

Subharmonics, modulations, and deterministic chaos in voice signals have been described for decades (scandinavian cry analysis since 1960, W. Mende et al. 1990...). They are described in newborn cries, voice pathologies, voice registers, singing voices, Russian lament (M. Mazo), and in numerous animal vocalizations. There are many speculations regarding possible functions of these nonlinear phenomena (encoding individuality, enhancing emotions...). Most papers discussing this issue are, however, somewhat speculative since statistical data on context, signal quantification, perceptual evaluation etc. are sparse.

The submitted manuscript addresses this long-standing open problem - possible functions of nonlinear phenomena - in a creative and comprehensive way. The experimental design is done very carefully, the statistics (model selection, WAIC, ..) is appropriate, and the sample sizes are impressive. The authors use a carefully designed set of raw and modified vocalizations. They find as a starting point strong effects of the context on the valence but not on perceived intensity. The effects of nonlinear phenomena (NLPs) are less pronounced but thanks to large sample size and appropriate statistical analysis interesting conclusions could be drawn: listeners associate NLPs with aversive states and the context could even be changed from positive to negative. The exciting results expand our view on nonverbal communication and might even be a starting point to study possible functions of NLPs in non-human vocalizations.

Review form: Reviewer 2

Is the manuscript scientifically sound in its present form?

Yes

Are the interpretations and conclusions justified by the results?

Yes

Is the language acceptable?

Yes

Do you have any ethical concerns with this paper?

No

Have you any concerns about statistical analyses in this paper?

No

Recommendation?

Accept with minor revision (please list in comments)

Comments to the Author(s)

This paper investigates the effect of non-linear phenomena (NLP, in particular sub-harmonics and chaos), whose function is still debated. Indeed, they are usually assumed to be associated with an increase in negative arousal (e.g. fear), although when comparing between species, these features do not predictively increase with arousal. The three experiments conducted in this study nicely show that NLPs mainly increase the perceived aversiveness of vocalisations, in both negative and positive contexts, and only to a smaller extent, their perceived emotion intensity. Controlled synthetic sounds were used and the experiments and analyses seem correct to me. This paper would thus be a very valuable and nice addition to the literature of vocal expression of emotions.

I only have a few comments to improve the clarity of the manuscript:

P2 L38-40. Pilot study. Was one synthetic sound build to match each of the 61 selected calls?

P2 L40-41. How were long bouts truncated (was the end part cut out, or the beginning)?

P2 L43. What is meant by "unsuitable" ratings?

P3 L42-43. 'The same images were shown in all trials in a block – that is, versions of the same prototype stimulus with and without NLPs were presented with the same (positive or negative) contextual cues'. This is not entirely clear. Do you mean that the same version of a stimulus with and without NLP was presented with the same image to one given participant?

P5 L28 - P6 L10. It would help the reader if the rationale behind each experiment (what information each of them brings) is explained briefly before describing the procedure, along with predictions.

P6 L33-36. What was the random effect?

Results. Throughout the results, what is regarded as no / moderate / strong effect? This could be specified beforehand in the method section. Different terms are used throughout the results (modest, marginal, small). It would be good to define how effect sizes are categorized. Also, could the WAIC results be added? (e.g. effect of the interaction between the main effects – justifying why you then look at the effect of call type, measure and manipulation -> were these interactions present in the best selected model?)

Decision letter (RSOS-201306.R0)

Dear Dr Anikin,

On behalf of the Editors, we are pleased to inform you that your Manuscript RSOS-201306 "Do nonlinear vocal phenomena signal negative valence or high emotion intensity?" has been accepted for publication in Royal Society Open Science subject to minor revision in accordance with the referees' reports. Please find the referees' comments along with any feedback from the Editors below my signature.

Please submit your revised manuscript and required files (see below) no later than 7 days from today's (ie 21-Oct-2020) date. Note: the ScholarOne system will 'lock' if submission of the revision is attempted 7 or more days after the deadline. If you do not think you will be able to meet this deadline please contact the editorial office immediately.

Please note article processing charges apply to papers accepted for publication in Royal Society Open Science (<https://royalsocietypublishing.org/rsos/charges>). Charges will also apply to papers transferred to the journal from other Royal Society Publishing journals, as well as papers submitted as part of our collaboration with the Royal Society of Chemistry

(<https://royalsocietypublishing.org/rsos/chemistry>). Fee waivers are available but must be requested when you submit your revision (<https://royalsocietypublishing.org/rsos/waivers>).

on behalf of Dr César Lima (Associate Editor) and Essi Viding (Subject Editor)
openscience@royalsociety.org

Reviewer comments to Author:

Reviewer: 1

Comments to the Author(s)

Subharmonics, modulations, and deterministic chaos in voice signals have been described for decades (scandinavian cry analysis since 1960, W. Mende et al. 1990...). They are described in newborn cries, voice pathologies, voice registers, singing voices, Russian lament (M. Mazo), and in numerous animal vocalizations. There are many speculations regarding possible functions of these nonlinear phenomena (encoding individuality, enhancing emotions...). Most papers discussing this issue are, however, somewhat speculative since statistical data on context, signal quantification, perceptual evaluation etc. are sparse.

The submitted manuscript addresses this long-standing open problem - possible functions of nonlinear phenomena - in a creative and comprehensive way. The experimental design is done very carefully, the statistics (model selection, WAIC, ..) is appropriate, and the sample sizes are impressive. The authors use a carefully designed set of raw and modified vocalizations. They find as a starting point strong effects of the context on the valence but not on perceived intensity. The effects of nonlinear phenomena (NLPs) are less pronounced but thanks to large sample size and appropriate statistical analysis interesting conclusions could be drawn: listeners associate NLPs with aversive states and the context could even be changed from positive to negative. The exciting results expand our view on nonverbal communication and might even be a starting point to study possible functions of NLPs in non-human vocalizations.

Reviewer: 2

Comments to the Author(s)

This paper investigates the effect of non-linear phenomena (NLP, in particular sub-harmonics and chaos), whose function is still debated. Indeed, they are usually assumed to be associated with an increase in negative arousal (e.g. fear), although when comparing between species, these features do not predictively increase with arousal. The three experiments conducted in this study nicely show that NLPs mainly increase the perceived aversiveness of vocalisations, in both negative and positive contexts, and only to a smaller extent, their perceived emotion intensity. Controlled synthetic sounds were used and the experiments and analyses seem correct to me. This paper would thus be a very valuable and nice addition to the literature of vocal expression of emotions.

I only have a few comments to improve the clarity of the manuscript:

P2 L38-40. Pilot study. Was one synthetic sound build to match each of the 61 selected calls?

P2 L40-41. How were long bouts truncated (was the end part cut out, or the beginning)?

P2 L43. What is meant by "unsuitable" ratings?

P3 L42-43. 'The same images were shown in all trials in a block – that is, versions of the same prototype stimulus with and without NLPs were presented with the same (positive or negative) contextual cues'. This is not entirely clear. Do you mean that the same version of a stimulus with and without NLP was presented with the same image to one given participant?

P5 L28 - P6 L10. It would help the reader if the rationale behind each experiment (what information each of them brings) is explained briefly before describing the procedure, along with predictions.

P6 L33-36. What was the random effect?

Results. Throughout the results, what is regarded as no / moderate / strong effect? This could be specified beforehand in the method section. Different terms are used throughout the results (modest, marginal, small). It would be good to define how effect sizes are categorized. Also, could the WAIC results be added? (e.g. effect of the interaction between the main effects – justifying why you then look at the effect of call type, measure and manipulation -> were these interactions present in the best selected model?)

===PREPARING YOUR MANUSCRIPT===

===PREPARING YOUR REVISION IN SCHOLARONE===

To revise your manuscript, log into <https://mc.manuscriptcentral.com/rsos> and enter your Author Centre - this may be accessed by clicking on "Author" in the dark toolbar at the top of the

page (just below the journal name). You will find your manuscript listed under "Manuscripts with Decisions". Under "Actions", click on "Create a Revision".

<https://royalsociety.org/journals/authors/author-guidelines/#supplementary-material> to include a suitable title and informative caption. An example of appropriate titling and captioning may be found at https://figshare.com/articles/Table_S2_from_Is_there_a_trade-off_between_peak_performance_and_performance_breadth_across_temperatures_for_aerobic_sc_ope_in_teleost_fishes_/3843624.

Author's Response to Decision Letter for (RSOS-201306.R0)

See Appendix A.

Decision letter (RSOS-201306.R1)

Dear Dr Anikin,

It is a pleasure to accept your manuscript entitled "Do nonlinear vocal phenomena signal negative valence or high emotion intensity?" in its current form for publication in Royal Society Open Science. The comments of the reviewer(s) who reviewed your manuscript are included at the foot of this letter.

on behalf of Dr César Lima (Associate Editor) and Essi Viding (Subject Editor)
openscience@royalsociety.org

Appendix A

We are grateful to both Reviewers for the positive evaluation of our work.

#####

REVIEWER 1

#####

There are no specific comments of Reviewer 1 to address.

#####

REVIEWER 2

#####

R2_1 P2 L38-40. Pilot study. Was one synthetic sound build to match each of the 61 selected calls?

RESPONSE: Yes. To avoid ambiguity, the text was changed to:

The 61 vocalisations in this pilot study were fully synthetic, but they were designed to be as similar as possible to the original recording and contained the same NLPs, if any.

R2_2 P2 L40-41. How were long bouts truncated (was the end part cut out, or the beginning)?

RESPONSE: The selection of stimuli was performed before the pilot experiment, which was designed to confirm that the chosen sounds were emotionally intense and ambiguous, so we felt justified in relying on intuition as well as previously collected data when making this initial selection. As a result, the sounds were selected and then synthesized without following formal criteria such as always taking the first or last x seconds, etc. For example, a 3.5 s roar of pain was synthesized to be shorter (1.2 s), while its intonation and voice quality were reproduced closely. Moans of pleasure in the original corpus often included long, multi-syllable bouts, in which case one or a few syllables were selected, regardless of their location in the bout (beginning, middle, or end).

R2_3 P2 L43. What is meant by "unsuitable" ratings?

RESPONSE: Changed to:

*Based on the results of this pilot study, 13 out of 61 sounds were excluded because (a) **participants often chose the "skip" option when rating them in either positive or negative scenarios**, or (b) there was a large difference in intensity ratings between the positive and negative scenarios.*

R2_4 P3 L42-43. 'The same images were shown in all trials in a block - that is, versions of the same prototype stimulus with and without NLPs were presented with the same (positive or negative) contextual cues'. This is not entirely clear. Do you mean that the same version of a stimulus with and without NLP was presented with the same image to one given participant?

RESPONSE: For clarity, we changed the text to:

Only the images shown in Table 2 were used, the same for all participants and in all trials in a block - that is, versions of the same prototype stimulus with and without NLPs were presented with the same (positive or negative) contextual cues, enabling us to test the within-stimulus effect of acoustic manipulation in positive and negative contexts.

R2_5 P5 L28 - P6 L10. It would help the reader if the rationale behind each experiment (what information each of them brings) is explained briefly before describing the procedure, along with predictions.

RESPONSE: Thank you for an excellent suggestion! Added text (first paragraph of Procedure):

*We performed three playback experiments on independent samples of a total of 900 participants. Figure 2 illustrates the design of each experiment. **The valence ratings obtained in neutral, positive, and negative contexts (Experiments 1 and 2) enabled us to verify the effectiveness of contextual cues in shifting the perception of these ambiguous vocalisations, as could be predicted based on previous research [23,24]. We were particularly interested in confirming the finding by Atias et al. [23] that contextual cues sometimes suffice not only to modulate, but even to reverse or "flip" the perceived valence of a vocalisation. In addition, Experiments 1 and 2 addressed the effect of NLPs on perceived valence. The intensity ratings obtained in positive and negative contexts (Experiment 3) made it possible to test whether the addition of NLPs boosted the intensity of any emotion (as predicted by the attention-grabbing account) or specifically increased the perceived intensity of only negative affective states (as predicted by the harsh-is-aversive account).***

R2_6 P6 L33-36. What was the random effect?

RESPONSE: The random effects are explained in the next paragraph: “*In addition, the effect of adding NLPs in each experiment was allowed to vary across stimuli by specifying the corresponding random slopes for each of 48 prototype sounds*”. The model structure, in *brms* syntax, was as follows:

```
model = brm(  
  bf(  
    response ~ emGroup * cond * measure + (1|id) + (1|file) + (cond * measure|sound),  
    zoi ~ emGroup * cond * measure,  
    coi ~ emGroup * cond * measure  
  ),  
  family = 'zero_one_inflated_beta',  
  prior = c(set_prior('logistic(0, 1)', class = 'b', dpar = 'coi'),  
            set_prior('logistic(0, 1)', class = 'b', dpar = 'zoi')),  
  data = df, cores = 4, chains = 4, warmup = 500, iter = 2000  
)
```

R2_7 Results. Throughout the results, what is regarded as no / moderate / strong effect? This could be specified beforehand in the method section. Different terms are used throughout the results (modest, marginal, small). It would be good to define how effect sizes are categorized.

RESPONSE: All effect sizes are reported in natural units – in this case, points on a rating scale (0 to 100). The conventions of small/medium/large effects sizes that are sometimes (controversially!) suggested for standardized scales such as Cohen’s *d* or Bayes Factors are (1) not applicable to natural units and (2) not really necessary when the reader can easily appreciate whether or not a change of valence of, say, 3 (95% CI [1, 8]) points on a 0...100 scale is large enough to be of practical importance. We provided descriptive terms to express our own evaluation of how “exciting” (as opposed to statistically robust or significant) we considered these effects, for ex.: “*other call types were rated as **considerably** more aversive*”, “*context had a **relatively muted** effect on the perceived intensity of affective vocalisations*”, “*the average intensity of delight was **only** 4.0% [-1.1, 9.0] above the average intensity of fear*”, “*making them **only negligibly** more aversive (2.9%, 95% CI [0.8, 5.1])*”, etc.

These subjective quantifications of effect sizes make no claims about statistical inference, which is predicated on the precision of inferential point estimates – namely, the credible intervals indicating the most likely (or “credible”) parameter values given the data, model, and prior beliefs. Unfortunately, the terminology for reporting the results of Bayesian analyses is not yet firmly established, compounded by the existence of different approaches to the interpretation of the “significance” of particular effects (e.g., based on Bayes Factors / whether or not CIs include zero / Regions of Practical Equivalence or ROPEs / information criteria / ...). With the exception of testing for interactions with information criteria (see R2_8), in this text we focused on reporting natural effect sizes and their precision instead of making binary yes-no decisions about the “significance” of the reported contrasts. So, for example, we write “*The effect of context on the perceived emotion intensity was also **too small and uncertain to detect** for gasps (a difference of 3.1% [-2.7, 9.5])*” – that is, not only is the effect size small in substantive terms, but the CI includes negative values, so the conclusion is that the effect is either unexcitingly weak or absent altogether. The same

conclusion can be made more confidently when the CI is narrow and centred on zero, e.g. “Notably, there was **no overall difference** in the aversiveness of NLPs in negative versus positive contexts (-0.8% [-3.3, 1.6] for subharmonics / sidebands, -1.4% [-3.7, 0.8] for chaos”. In contrast, an effect may be small in substantive terms, but definitely non-zero, e.g. “this difference, **while relatively small, was robust statistically**: the addition of chaos enhanced the perceived emotion intensity by an additional 7.6% [5.0, 10.3]”.

We realize that this is a bit of a Wild West terminologically, and readers unfamiliar with Bayesian inference may be misled by terms similar to Null Hypothesis Significance Testing (NHST). We have therefore removed the word “marginally” from the text and included a reference to an excellent article-length introduction to the topic: *Kruschke, J. K., & Liddell, T. M. (2018). The Bayesian New Statistics: Hypothesis testing, estimation, meta-analysis, and power analysis from a Bayesian perspective. Psychonomic Bulletin & Review, 25(1), 178-206.*

R2_8 Also, could the WAIC results be added? (e.g. effect of the interaction between the main effects - justifying why you then look at the effect of call type, measure and manipulation -> were these interactions present in the best selected model?)

RESPONSE: In general, this is a common but statistically questionable assumption that the “true” model is always the simplest one, and “non-significant” interaction effects should be excluded even if they are of theoretical interest (see Ch. 6 of McElreath 2015 “Statistical rethinking: a Bayesian course” for a great discussion of these issues). In this case, “measure” represents different experiments with different rating scales, “call type” corresponds to different types of stimuli AND different contextual cues (different vignettes and images), while “manipulation” is what the study was designed to analyse. Accordingly, it would be strange to even consider models that do not include them, and the results also bear out that NLPs have a much stronger effect on valence without vs. with contextual cues, that context has different effects on the perceived intensity of moans vs. other vocalisations, etc. It must also be pointed out that a more conventional approach would be to build a separate model for each experiment and each scale (thus obviating the need for interaction effects), while we analysed this very large dataset (~40K datapoints from 900 participants) in one go.